# Educational attainment and differences in relative survival after acute myocardial infarction in Norway: a registry-based population study

Søren Toksvig Klitkou, Knut R Wangen

► Prepublication history and additional material are available. To view these files please visit the journal online (http://dx.doi.org/10.1136/bmjopen-2016-014787).

## ABSTRACT

**Background** Although there is a broad societal interest in socioeconomic differences in survival after an acute myocardial infarction, only a few studies have investigated how such differences relate to the survival in general population groups. We aimed to investigate education-specific survival after acute myocardial infarction and to compare this with the survival of corresponding groups in the general population.

**Methods** Our study included the entire population of Norwegian patients admitted to hospitals for acute myocardial infarction during 2008–2010, with a 6-year follow-up period. Patient survival was measured relative to the expected survival in the general population for three educational groups: primary, secondary and tertiary. Education, sex, age and calendar year-specific expected survival were obtained from population life tables and adjusted for the presence of infarction-related mortality.

**Results** Six-year patient survivals were 56.3% (55.3–57.2) and 65.5% (65.6–69.3) for the primary and tertiary educational groups (95% CIs), respectively. Also 6-year relative survival was markedly lower for the primary educational group: 70.2% (68.6–71.8) versus 81.2% (77.4–84.4). Throughout the follow-up period, patient survival tended to remain lower than the survival in the general population with the same educational background.

**Conclusion** Both patient survival and relative survival after acute myocardial infarction are positively associated with educational level. Our findings may suggest that secondary prevention has been more effective for the highly educated.

## INTRODUCTION

Socioeconomic differences in survival after acute myocardial infarction (AMI) are of broad societal interest, especially because ischaemic heart disease is the single most important cause of death, life-years lost prematurely and life-years lost to disability.[1 2] The causes of heart disease and AMI continue to be associated with lifestyles[3 4] and life conditions,[5–9] resulting in substantial socioeconomic differences in the rates of disease onset and mortality, which correlate with measures such as education level, income

and occupational class.[10–12] Inequalities in mortality have widened in both absolute and relative terms, a public health phenomenon that has been described as a great disappointment in highly developed welfare states.[13] This widening inequality is in large part attributable to the persistent relationship between socioeconomic status and mortality from cardiovascular disease.

## Strengths and limitations of this study

► The relative survival framework made possible the comparison of patient survival to population survival for a national cohort of patients with acute myocardial infarction (AMI) and to distinguish how groups defined by educational level fared compared with their group in the population.

► The use of nationwide registry data from patients and the general population give findings that are representative for Norway for the years of diagnosis from 2008 to 2010 with a maximum follow-up of 6 years until the end of 2013.

► The study has low levels of missing data on education (<2%) and identical definitions of educational attainment in patient and population groups.

Patient survival following AMI is commonly assessed using overall patient survival.[14] This can be appropriate from a clinical perspective, say when comparing treatment strategies. However, AMI occurs more frequently in age groups with elevated background mortality and it is therefore of interest to measure patient survival relative to the survival of the general population. In relative survival, this comparison is accomplished by life tables for population survival.[15]

Dickman *et al* emphasised the importance of stratifying the general population

Department of Health Management and Health Economics, Institute of Health and Society, University of Oslo, Oslo, Norway

**Correspondence to**
Søren Toksvig Klitkou;
s.t.klitkou@medisin.uio.no

life tables according to the adopted measure of socioeconomic status when the aim is to compare separate social groups.[16] The relative survival approach was later refined to situations where the disease of interest had a substantial influence on population survival—typically when a relatively common disease causes high mortality.[17] The principle of the correction was to adjust the population survival to resemble survival in a population *without* the disease of interest.

Socioeconomic and demographic gradients in relative survival have been studied previously for patients with AMI. A Singaporean study found lower relative survival among the Malayan minority, as compared with the Chinese and Indian ethnic groups,[18] while a Spanish study reported no association between relative survival and a social deprivation index.[19] However, neither these nor similar studies of cardiovascular disease applied the mentioned refined approach.[18–22]

In the context of this prior research, our aim was to investigate relative survival ratios after AMI for patients with different educational backgrounds in Norway. We used registry data that covered the entire Norwegian population. A single-payer healthcare system and a tuition-free higher education system are provided by the public sector in Norway. In principle, this offers the whole population equal access to healthcare and education. Educational inequalities in 28-day and 1-year AMI survival have been reported previously,[23] but no comparison with the general population has been published to date. Additionally, in contrast to previous studies on relative survival for cardiovascular disease,[18–22] we have accounted for AMI mortality in the general population.

## METHODS
### Data
For the current study, we considered all hospital admissions for AMI (International Classification of Diseases (ICD-10) I21 or I22 as main diagnosis) that were included in the Norwegian National Patient Registry for the period 2008–2010. Information on patient survival was derived from date of death, from the Causes of Death Registry for the period 2008–2013. This registry covers all deaths in Norway and deaths of Norwegians who die abroad. Data on educational level were obtained from the National Database on Education.

We identified 35 045 patients aged 40–94 years with an AMI diagnosis in the patient registry. Patients who were younger or older at the time of infarction were excluded because of small numbers of patients in these groups. For patients with multiple AMI episodes during the observation period (2008–2010), only the first episode was included in the analysis. Information on AMI episodes prior to the observation period was not available. Educational level was defined as the highest completed level of education, and was categorised as follows: primary (6–9 years of schooling), secondary (10–12 years of schooling) or tertiary (university or college degree). We excluded

637 patients with missing information on attained education, leaving a total sample of 34 408 patients. In all 12 135 persons died during the course of follow-up. The observed number of person-years was 116 269, with an average follow-up of 3.4 years per person.

To calculate relative survival ratios, we obtained life tables from Statistics Norway for the calendar years 1999–2010, which included data by educational level, sex and 1-year age interval. The life tables included all deaths and person-years, permitting the calculation of expected survival. Importantly, for our application, the patients and the general population were stratified using identical definitions of educational groups.[16 24]

### Statistical analyses
For a given calendar year and population stratum (educational level, sex and age), relative survival was defined as the ratio of survival among the patients with AMI to the expected survival for the general population.

The expected survival was estimated in two steps. In the first step, overall expected survival for the years 2008–2010 was obtained directly from the life tables covering the years 1999–2010. Data for the years 2011–2013 were not available and were thus extrapolated based on the 1999–2010 data. For this purpose, we estimated six Poisson regression models—one for each combination of educational level and sex. The dependent variable was the number of deaths per person-year, observed for each 1-year age category (40–94 years) and for each calendar year (1999–2010). The independent variables included a linear trend for calendar year and restricted cubic splines to represent a possible non-linear age effect,[25] where the number of knots used was decided by Akaike Information Criterion. The estimated models were then used to predict values for 2011–2013. As a sensitivity analysis, we used two alternative approaches to obtain values for 2011–2013: one approach suggested by Dickman *et al* and explained in online supplementary description S1[16] used standardised mortality ratios by educational level for the years 2008–2010 to adjust Norwegian mortality ratios available from the Human Mortality Database for the years 2008–2013.[26] A second approach consisted of simply forwarding the estimates from the last available year of our life table data (ie, 2010). In the second step, we adjusted the overall expected survival for the substantial proportion of deaths due to AMI, by using Talbäck and Dickman's correction formula.[17] To this end, we obtained data from the Norwegian Institute of Public Health on the sex and age group-specific proportions of deaths due to AMI.[27] As these data did not include educational level, we performed a sensitivity analysis based on published comparisons of cardiovascular mortality (ICD-10 I00–I99) with overall mortality (see online supplementary description S2).[28]

We summarised survival differences by level of education at the end of follow-up, as stratified by sex and 5-year age groups. To investigate the time course of survival since AMI, we calculated survival for the first month and

each year thereafter. For each of these time points, we calculated relative survival (using the Ederer II method) by directly standardising the three educational groups to the overall age and sex distribution.[25 29] The standardised survival was defined as the weighted average of the survival in each age and sex group, where the weights equalled the proportions of patients in each group. Finally, we performed a continuous-time analysis of survival, in which the time course of survival since AMI was estimated using the Kaplan-Meier and the time course of expected survival was estimated using the Ederer I method,[25] as standardised by reweighting to the overall age and sex distribution. Stata V.14 and the Strs-procedure were used for the calculations in the tables.[25] Additionally, we used R V. 3.1.2 and the Survival package for continuous-time estimates.

## RESULTS

The average age at hospitalisation was 71 years, and 36% of the patients were female. Of the 34 408 patients, 14 360 had completed primary education, 15 527 had completed secondary education and 4521 had completed tertiary education.

Table 1 shows the survival estimates, stratified by sex, education and 5-year age groups: patient survival at 6 years of follow-up; the expected survival if the patients had experienced the mortality rates of the general population; and the relative survival, defined as the ratio of the patient survival to the expected survival. The expected survival is obtained from the full population and has a clear, consistent pattern: for a given sex and educational group, it is decreasing in age; for a given age and educational level, it is higher for females than for males, and; for a given age and sex, it is increasing by educational level. Patient survival and relative survival also vary but their patterns appear to be more consistent for males (n=21 878) than for females (n=12 530), presumably due to the higher number of male patients. For males above 55 years, patient survival clearly differs according to educational level. For example, there is a notable difference for the age group 55–59 years, where the respective patient survivals in the primary, secondary and tertiary educational groups were 80.9%, 87.1% and 92.5%. Dividing by the corresponding expected survivals, this translates into the relative survivals of 85.8%, 90.1% and 94.4%, for the primary, secondary and tertiary educational groups, respectively. A higher relative survival indicates that the patient survival remained closer to the survival among members of the general population background. For females, the relative survival also has a similar socioeconomic gradient, except for the age groups 50–54, 55–59 and 80–84, but the 95% CIs are rather wide.

The results in tables 2 and 3 are standardised to the combined age and sex distribution of all the observed patients. Table 2 presents the time course of patient survival. The 6-year survival was 56.3% and 67.5% for patients with primary and tertiary education, respectively.

Accounting for the expected survival in the general population, Table 3 presents the time course of relative survival ratios over the follow-up period. In each year, the patients with AMI fared worse than their counterparts in the general population. Survival decreased substantially during the first year since AMI, for which the relative survival ratios were 83.0%, 84.6% and 87.0% in the primary, secondary and tertiary education groups, respectively. By year 6, ratios of relative survival were 70.2%, 73.1% and 81.2% in these groups—the difference in relative survival had become markedly larger. For the primary and secondary education groups, the decline in relative survival was monotonic over the 6-year period. In contrast, for the tertiary education group, the relative survival ratio appeared to stabilise at approximately 81% after 5 years; however, the statistical basis of this is uncertain.

The patterns observed in table 2 can be explored further by comparing patient survival and expected survival on a continuous time scale (figure 1). For the primary and secondary education groups, the patient survival curve has a consistently steeper slope than the expected survival curve. In contrast, these curves become nearly parallel in some periods for the tertiary education group.

Summarising the findings from figure 1 and table 2, we observed that differences in overall, expected and relative survival all appeared to increase with time since AMI to the benefit of patients with higher education levels. The findings of the previously specified sensitivity analyses did not differ meaningfully from the results presented above.

## DISCUSSION

The purpose of this study was to compare patient survival with population survival across levels of education, which was used as a marker of socioeconomic status. Our findings confirm previous studies that have documented large and continuing differences in patient survival (see Osler *et al* for a recent overview).[12] Additionally, our analysis demonstrates how these differences in post-AMI survival relate to the expected survival of different educational groups in the general population. According to our results, the relative survival of patients with a primary level of education corresponded to those of patients with a tertiary level of education who are 5 to 10 years older. The main result is that the survival of patients with a tertiary education level remained closest to the survival seen for members of the general population with the same educational background. Given the relation of socioeconomic status to general population mortality,[4 13] we expect that the current result would be replicable in other countries and for other diseases exhibiting a socioeconomic dimension to survival.

A core strength of our study is the use of educational and patient data on the individual level for the entire Norwegian population. This was possible because the Norwegian National Patient Registry was made linkable in 2008, with the tripartite aim of safeguarding access to

**Table 1** The 6-year survival rates of patients after acute myocardial infarction, according to sex, age and education level. Observed patient survival, expected survival[1] and relative survival ratios are shown.

**Males**

| Age, years | Primary education | | | | | | Secondary education | | | | | | Tertiary education | | | | | |
|---|---|---|---|---|---|---|---|---|---|---|---|---|---|---|---|---|---|---|
| | n | PS (95% CI) | | ES | RS (95% CI) | | n | PS (95% CI) | | ES | RS (95% CI) | | n | PS (95% CI) | | ES | RS (95% CI) | |
| 40–49 | 702 | 90.0 | (86.5 to 92.6) | 98.2 | 91.7 | (88.1 to 94.4) | 814 | 93.6 | (90.8 to 95.6) | 99.0 | 94.5 | (91.7 to 96.5) | 316 | 89.2 | (83.4 to 93.1) | 99.5 | 89.7 | (83.9 to 93.6) |
| 50–54 | 531 | 88.0 | (83.6 to 91.3) | 96.4 | 91.2 | (86.7 to 94.6) | 985 | 89.2 | (86.3 to 91.6) | 98.0 | 91.1 | (88.1 to 93.5) | 317 | 91.5 | (86.1 to 94.9) | 98.9 | 92.6 | (87.1 to 96.0) |
| 55–59 | 615 | 80.9 | (76.3 to 84.7) | 94.2 | 85.8 | (81.0 to 89.9) | 1310 | 87.1 | (84.4 to 89.4) | 96.7 | 90.1 | (87.3 to 92.5) | 461 | 92.5 | (88.3 to 95.2) | 98.0 | 94.4 | (90.2 to 97.2) |
| 60–64 | 819 | 74.6 | (70.4 to 78.3) | 92.0 | 81.1 | (76.6 to 85.1) | 1765 | 79.9 | (77.3 to 82.3) | 94.5 | 84.6 | (81.8 to 87.1) | 583 | 86.3 | (81.9 to 89.6) | 96.7 | 89.2 | (84.7 to 92.7) |
| 65–69 | 794 | 63.2 | (58.8 to 67.3) | 87.9 | 71.9 | (66.9 to 76.6) | 1294 | 72.7 | (69.3 to 75.7) | 91.5 | 79.4 | (75.8 to 82.8) | 495 | 81.7 | (76.5 to 85.8) | 94.2 | 86.7 | (81.3 to 91.1) |
| 70–74 | 870 | 54.5 | (50.3 to 58.5) | 82.5 | 66.1 | (61.0 to 71.0) | 1148 | 59.7 | (56.0 to 63.2) | 86.6 | 68.9 | (64.7 to 73.0) | 366 | 67.9 | (61.3 to 73.6) | 90.7 | 74.9 | (67.5 to 81.2) |
| 75–79 | 1020 | 40.4 | (36.8 to 44.0) | 72.4 | 55.8 | (50.9 to 60.7) | 1108 | 43.4 | (39.9 to 46.9) | 76.5 | 56.8 | (52.2 to 61.3) | 319 | 54.9 | (47.8 to 61.4) | 81.6 | 67.3 | (58.6 to 75.2) |
| 80–84 | 1124 | 23.9 | (21.1 to 26.7) | 56.2 | 42.4 | (37.6 to 47.4) | 1096 | 27.1 | (24.2 to 30.1) | 62.4 | 43.4 | (38.7 to 48.2) | 299 | 31.1 | (25.2 to 37.2) | 68.6 | 45.4 | (36.8 to 54.3) |
| 85–89 | 846 | 11.4 | (9.3 to 13.8) | 38.5 | 29.6 | (24.1 to 35.8) | 863 | 12.7 | (10.5 to 15.2) | 42.3 | 30.1 | (24.8 to 36.0) | 232 | 24.4 | (18.5 to 30.8) | 50.0 | 48.8 | (37.0 to 61.5) |
| 90–94 | 363 | 5.1 | (3.1 to 7.7) | 21.4 | 23.7 | (14.5 to 36.2) | 345 | 6.0 | (3.8 to 8.9) | 22.8 | 26.3 | (16.5 to 39.2) | 77 | 13.2 | (6.5 to 22.4) | 28.5 | 46.4 | (22.8 to 78.5) |

**Females**

| Age, years | Primary education | | | | | | Secondary education | | | | | | Tertiary education | | | | | |
|---|---|---|---|---|---|---|---|---|---|---|---|---|---|---|---|---|---|---|
| | n | PS (95% CI) | | ES | RS (95% CI) | | n | PS (95% CI) | | ES | RS (95% CI) | | n | PS (95% CI) | | ES | RS (95% CI) | |
| 40–49 | 170 | 83.8 | (74.6 to 89.9) | 98.9 | 84.8 | (75.4 to 90.9) | 152 | 88.8 | (79.6 to 94.0) | 99.4 | 89.4 | (80.1 to 94.6) | 80 | 90.5 | (76.6 to 96.3) | 99.6 | 90.9 | (76.9 to 96.7) |
| 50–54 | 162 | 90.6 | (82.1 to 95.2) | 97.7 | 92.8 | (84.0 to 97.5) | 176 | 84.3 | (75.3 to 90.2) | 98.6 | 85.5 | (76.3 to 91.5) | 83 | 88.6 | (74.8 to 95.1) | 98.9 | 89.6 | (75.7 to 96.2) |
| 55–59 | 177 | 89.3 | (81.0 to 94.1) | 96.6 | 92.4 | (83.9 to 97.4) | 311 | 89.1 | (83.2 to 93.0) | 97.9 | 90.9 | (85.0 to 94.9) | 89 | 97.8 | (85.3 to 99.7) | 98.6 | 99.2 | (86.5 to 101.1) |
| 60–64 | 360 | 69.8 | (63.1 to 75.5) | 95.0 | 73.5 | (66.5 to 79.5) | 434 | 79.7 | (74.1 to 84.3) | 96.5 | 82.6 | (76.7 to 87.3) | 121 | 86.2 | (75.1 to 92.5) | 97.8 | 88.1 | (76.8 to 94.6) |
| 65–69 | 427 | 65.2 | (59.0 to 70.6) | 92.6 | 70.4 | (63.8 to 76.3) | 418 | 70.6 | (64.5 to 75.9) | 95.0 | 74.3 | (67.9 to 79.9) | 102 | 87.2 | (74.9 to 93.7) | 96.5 | 90.3 | (77.7 to 97.1) |
| 70–74 | 644 | 55.0 | (50.1 to 59.6) | 88.9 | 61.9 | (56.3 to 67.1) | 501 | 61.6 | (56.0 to 66.8) | 92.4 | 66.7 | (60.6 to 72.3) | 108 | 67.4 | (54.6 to 77.4) | 94.1 | 71.7 | (58.0 to 82.3) |
| 75–79 | 942 | 41.1 | (37.4 to 44.8) | 82.1 | 50.1 | (45.5 to 54.6) | 577 | 48.5 | (43.5 to 53.4) | 86.2 | 56.3 | (50.4 to 61.9) | 119 | 64.1 | (52.0 to 74.0) | 89.1 | 72.0 | (58.4 to 83.0) |
| 80–84 | 1376 | 24.5 | (22.0 to 27.0) | 68.9 | 35.5 | (31.9 to 39.2) | 834 | 31.3 | (27.8 to 35.0) | 74.3 | 42.2 | (37.4 to 47.1) | 125 | 32.3 | (23.1 to 41.8) | 78.6 | 41.1 | (29.4 to 53.2) |
| 85–89 | 1548 | 13.8 | (12.1 to 15.7) | 51.2 | 27.0 | (23.5 to 30.7) | 864 | 18.4 | (15.6 to 21.3) | 56.7 | 32.4 | (27.6 to 37.5) | 146 | 22.7 | (15.7 to 30.5) | 59.7 | 38.0 | (26.2 to 51.2) |
| 90–94 | 870 | 8.4 | (6.6 to 10.5) | 31.0 | 27.1 | (21.3 to 33.7) | 531 | 9.7 | (7.3 to 12.6) | 34.5 | 28.1 | (21.1 to 36.3) | 83 | 14.5 | (7.6 to 23.5) | 41.1 | 35.2 | (18.4 to 57.3) |

[1]Expected survival has been corrected for mortality from acute myocardial infarction (AMI). Measurement errors in expected survival were ignored. The patient survival estimates are by year six and were based on the subsample of patients diagnosed with AMI in 2008 with complete follow-up.
ES, expected survival; PS, patient survival; RS, relative survival.

**Table 2** Age and sex-standardised patient survival, by period and educational group (n=34 408)

| Period | Primary education Patient survival | 95% CI | Secondary education Patient survival | 95% CI | Tertiary education Patient survival | 95% CI |
|---|---|---|---|---|---|---|
| 1 month | 89.2 | 88.7 to 89.7 | 90.7 | 90.2 to 91.2 | 91.5 | 90.5 to 92.5 |
| Year 1 | 79.6 | 79.0 to 80.2 | 81.7 | 81.1 to 82.3 | 84.5 | 83.2 to 85.6 |
| Year 2 | 73.6 | 73.0 to 74.2 | 76.5 | 75.9 to 77.1 | 80.4 | 79.1 to 81.6 |
| Year 3 | 68.5 | 67.8 to 69.1 | 71.7 | 71.1 to 72.4 | 77.0 | 75.6 to 78.2 |
| Year 4 | 63.8 | 63.2 to 64.5 | 67.6 | 67.0 to 68.3 | 73.2 | 71.8 to 74.5 |
| Year 5 | 59.9 | 59.2 to 60.7 | 64.1 | 63.4 to 64.8 | 69.5 | 68.0 to 71.0 |
| Year 6 | 56.3 | 55.3 to 57.2 | 60.5 | 59.5 to 61.4 | 67.5 | 65.6 to 69.3 |

healthcare, ensuring appropriate quality of services, and monitoring disease occurrence in the population.

Relative survival has been suggested as a method for monitoring cardiovascular diseases,[14] but few applications have been published.[18–22] Our work appears to be the first study in which expected survival has been adjusted for infarction-related mortality—and this needs to be accounted for when making comparisons. Ideally, patient survival should be viewed in comparison with the expected survival that would be seen in a comparable population that is free of the disease in question. Survival in the general population is often used as an approximate estimate of expected survival, but because deaths among patients with AMI constitute a substantial share of the general population's mortality, this approximation can lead to underestimation of expected survival.[17] Hence, if we had not adjusted the expected survival, our estimates of relative survival would have been higher.

Although we have studied relative survival by education, the same methodology may also facilitate comparisons across countries or over time.[21 30] Where the comparison is of countries that have different life expectancies or coding practices for causes of death, the use of relative survival provides an external adjustment for survival differences that are unrelated to the disease under study. Therefore, relative survival ratios can help to reveal how survival is affected by international differences in disease characteristics and treatments.

A previous, population-based study reported that survival among elderly patients with AMI who underwent percutaneous coronary intervention (PCI) was comparable to the general population after about 6 months, and remained slightly above that of the general population for the remaining 3-year follow-up.[20] These estimates of relative survival are higher than our own, which may partly be due to the mentioned difference in the expected survival calculations, but may also be attributable to the fact that patients who underwent PCI tend to have higher survival than patients with AMI, in general.

We have found only two previous studies which have related relative survival of patients with AMI to socioeconomic factors.[18 19] For patients with AMI who survived the first 24 hours, a Singaporean study reported 5-year relative survival ratios of 69%, 73% and 79%, for Malayan, Chinese and Indian ethnic groups, respectively.[18] A Spanish study classified patients with AMI using a social deprivation index based on census tract,[19] but found no association between relative survival and deprivation for men. For women, the association was significant but, surprisingly, patients with intermediate deprivation levels had higher relative survival than the least deprived.

We used attained educational level as a measure of individual socioeconomic status. The educational level is usually attained in young adulthood when populations are overall in good health, and determined before later life social aspects such as income or occupation.[31] Groups

**Table 3** Age and sex-standardised relative survival, by period and educational group (n=34 408)

| Period | Primary education Relative survival | 95% CI | Secondary education Relative survival | 95% CI | Tertiary education Relative survival | 95% CI |
|---|---|---|---|---|---|---|
| 1 month | 89.6 | 89.1 to 90.0 | 91.0 | 90.5 to 91.5 | 91.8 | 90.7 to 92.7 |
| Year 1 | 83.0 | 82.3 to 83.6 | 84.6 | 84.0 to 85.3 | 87.0 | 85.6 to 88.2 |
| Year 2 | 79.8 | 79.0 to 80.5 | 81.9 | 81.2 to 82.7 | 85.2 | 83.7 to 86.7 |
| Year 3 | 77.1 | 76.3 to 77.9 | 79.4 | 78.5 to 80.2 | 84.3 | 82.5 to 85.9 |
| Year 4 | 74.4 | 73.5 to 75.3 | 77.2 | 76.2 to 78.1 | 82.5 | 80.4 to 84.3 |
| Year 5 | 72.4 | 71.3 to 73.5 | 75.4 | 74.3 to 76.5 | 80.8 | 78.3 to 83.1 |
| Year 6 | 70.2 | 68.6 to 71.8 | 73.1 | 71.4 to 74.7 | 81.2 | 77.4 to 84.4 |

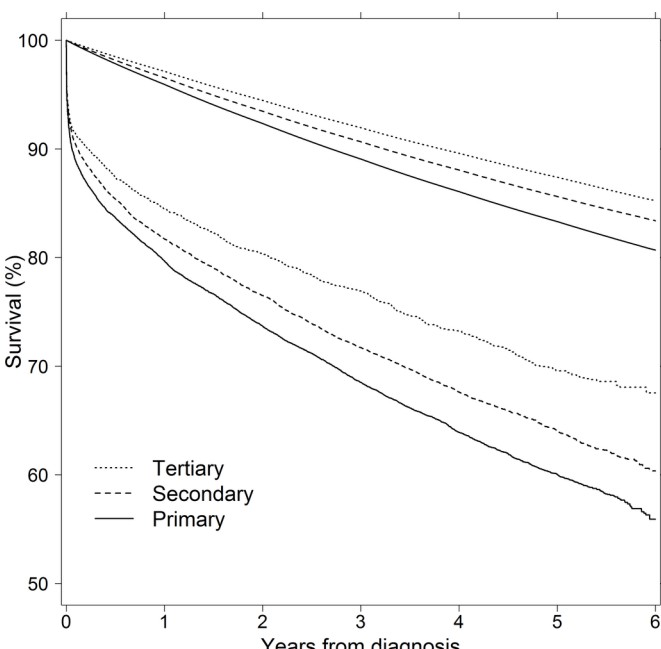

**Figure 1** Patient survival and expected survival, according to years since diagnosis and educational group. The upper, nearly linear curves show expected survival rates, as calculated using the Ederer I method. The lower curves show patient survival rates, as estimated using the Kaplan-Meier method. Results are standardised for age and sex.

with lower education levels are likely to experience more severe infarctions, owing to for instance more comorbidities and the presence of risk factor accumulation.[32] After an infarction, groups with lower educational attainment will in all likelihood be in command of significantly fewer resources to mobilise,[33 34] a situation that affects both clinical outcomes and lifestyle changes.[35–37] Each of these factors is only partially within the direct control of healthcare services, whose mandate is to base choice of treatment on the existing health and comorbidities of the patient. The appropriate channel of influence for factors affecting survival therefore also lies with the overall society. All else equal, greater infarction severity and more limited resources may therefore extend the period that patients with low education levels need to recover, thereby leading to reductions in survival that persist for a longer period. Similar mechanisms could apply for other causes of mortality—such as strokes, some types of cancers and accidents—which may explain the educational differences in expected survival.[28] Also, both patient and relative survival had clear educational differences in the aggregate (tables 2 and 3), while the differences were less clear for some, particularly younger age groups (table 1). It is beyond our scope to investigate this further, but it could be due to factors not accounted for, such as differences by birth cohorts in the advantage education confers,[31] or due to the relatively few observations in the disaggregated groups.

There are some limitations to the study. First, our data do not contain information on AMI episodes prior to

the observation period (2008–2010). Thus, our measurements of patient survival were obtained from a group where some patients had their first AMI episode while others may have had two or more. Previous studies that have reported survival after the first AMI episode are relevant for comparison.[12] However, it should be noted that if patient survival is generally higher after the first AMI episode than after later AMI episodes, our results would tend to be lower than for a sample with first AMI episode patients only. This effect of mixing patients may not have uniform size across educational groups, partly because educational differences affect the incidence of AMI and because their survival after an AMI differ, as demonstrated by our results.[11]

The second limitation is our reliance on at least partially modelled estimates for expected survival. We investigated the plausibility of our results in two sensitivity analyses: one sensitivity analysis relied on the latest available estimates from 2010. The other sensitivity analysis used standardised mortality ratios for the years 2008–2010, and applying these ratios to year 2008–2013 life tables from the Human Mortality Database, as suggested by Dickman et al[16] and implemented by Eleoranta et al.[38] We did not find any notable differences in our results when we applied these alternative estimates of expected survival.

A third limitation is that we were unable to provide separate corrections of the life tables for infarction-related mortality in each of the three education groups. As a sensitivity analysis, we therefore approximated such a correction by setting any difference equal to the relationship for overall cardiovascular mortality.[28] This adjustment is only appropriate to the extent that the relationship between educational level and overall cardiovascular disease is representative of AMI. We believe this to be the case. The application of this sensitivity analysis did not lead to differences from our previously obtained results, perhaps because cardiovascular mortality is responsible for about the same proportion of deaths, even in education groups with quite different overall mortality.[12 28 39]

These limitations notwithstanding, we believe that there are clear recommendations to be made on the basis of this study, which may inform models that compare the health benefits and cost consequences of cardiovascular disease interventions.[40] Thus, relevant cost-effectiveness analyses should incorporate the notion of lower relative survival among patients with AMI, which extends beyond the first year after AMI (as is common). Furthermore, where appropriate, models should incorporate relationships with received treatment, as in the study by Velders et al.[20]

In our opinion, including expected survival for patients with AMI and considering relative survival could provide a complementary estimate of the space in which improvements to patient survival can be made. Regarding our comparisons across levels of socioeconomic status, our findings suggest that both clinically and societally meaningful increases in the number of years lived could be obtained if the relative survival of patients with AMI

with lower educational levels was increased to the levels observed for patients with higher educational levels.

**Acknowledgements** The authors are grateful to the Norwegian Patient Registry and Statistics Norway for allowing us to use their data. The authors thank Tron Anders Moger and Tor Iversen for commenting on the study. The authors also thank the reviewers for providing constructive comments that helped clarify and improve the paper.

**Contributors** STK conceived the study, managed the data, performed the statistical analyses, and wrote the first draft. KRW participated in the interpretation of the statistical analyses and in redrafting the manuscript. Both authors were critically important for the manuscript's intellectual content and have approved its final version.

**Funding** The establishment of the research database was partially funded by the European Union 7th Framework Programme, contract no. 241721,(the EuroHOPE project).

**Competing interests** None declared.

**Patient consent** Registry-based study. Under the approval from the ethics Committee we are exempted to ask for consent from patients.

**Ethics approval** Regional Committees for Medical and Health Research Ethics (REC South East Norway), project no. 2009/2114.

**Provenance and peer review** Not commissioned; externally peer reviewed.

**Data sharing statement** Due to legal restrictions, we are prevented from sharing individual-level data. Researchers who wish to access data from the Norwegian Directorate of Health (the Norwegian Patient Registry) and Statistics Norway are advised to contact these institutions directly.

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
