## [Reviewer comments · BMJ Open]

ARTICLE DETAILS

TITLE (PROVISIONAL)	Educational attainment and differences in relative survival after acute myocardial infarction in Norway: a registry-based population study
AUTHORS	Klitkou, Søren; Wangen, Knut

VERSION 1 - REVIEW

REVIEWER	Sean Randall Curtin University, Australia
REVIEW RETURNED	07-Dec-2016

GENERAL COMMENTS	A pleasure to read.
---------------------

REVIEWER	Corrado De Vito Sapienza University of Rome
REVIEW RETURNED	13-Dec-2016

GENERAL COMMENTS	The manuscript is a well-written paper that assesses the differences in relative survival after myocardial infarction associated to educational level. However, I find the study an excellent exercise of biostatistics but poorly understood by a wide audience with a medium level of statistical knowledge. Honestly, the methods section cannot be understood by general readership in many parts and, finally, this great amount of work does not produce results so egregious as to modify the current knowledge in the field of inequalities. It is strongly recommended to send the manuscript to a more appropriate journal focused on epidemiologic and biostatistics arguments.
---

REVIEWER	Isidora Vujcic Institute of Epidemiology, Faculty of Medicine, University of Belgrade, Serbia
REVIEW RETURNED	18-Jan-2017

GENERAL COMMENTS	Comments to the Author Review of the paper "Educational attainment and differences in relative survival after acute myocardial infarction in Norway" This paper is concerned with investigating education-specific survival rates after acute myocardial infarction and comparing these rates with survival of corresponding groups in the general population. It is huge study conducted on 34,408 patients with an
---

	average follow-up of 3.4 years per person. I really enjoyed reading the paper. I have few questions for authors. 1. In results section authors mentioned that there are no indications of a socioeconomic gradient to the differences in relative survival for the age groups 40-49 and 50-54 years and, if anything, the pattern is mixed in these groups. How these findings can be explained? It suggest that AMI patient survival in younger age groups do not depend on education level? 2. Also, authors reported that expected survival rate is always higher for higher levels of education but without explanation. What are possible reasons for higher expected survival rate for higher levels of education, or opposite what are possible reasons for lower expected survival rate for lower levels of education? 3. Survival of AMI patients depend on many factors, not only on education. Is education independently associated with survival after AMI? Did treatment of AMI patients differ between educational groups? Is it possible, for instance that AMI patients with higher education get better medical treatment? Is there possibility for authors to adjust for comorbidities, revascularization or localization of MI? It should be discussed thoroughly and mentioned in study limitations
--	--

REVIEWER	Annette Kjær Ersbøll University of Southern Denmark United Kingdom of Great Britain and Northern Ireland
REVIEW RETURNED	19-Feb-2017

GENERAL COMMENTS	The authors use the first AMI in the study period if an individual had more than AMI. However, it is not clear from the description, if the authors are modelling incident AMI (i.e. first ever AMI) or first in the study period. Did authors use only main hospital diagnosis or all diagnoses including supplementary diagnoses? Due to lack of survival data for the years 2011-13 model-based predicted survival was used for the analysis, where the prediction was performed based on age, sex and education. This is considered a limitation. The research objective is relevant, however, data are not available to support this analysis As far as I understand data before 2008 in the Norwegian Patient Register was de-identified. Therefore, before 2008 data from the Norwegian National Patient Register could not be linked with other data sources. This is a limitation of the study as incident AMI in the study period (from 2010) is only possible to identify based on previous AMI's in the period 2008-9. Authors refer to results in table 1 as rates, however, it seems to be proportions and not rates.
--

REVIEWER	Patricia Morton Rice University USA
REVIEW RETURNED	15-May-2017

GENERAL COMMENTS**Methodological Review**

This paper utilizes novel data to investigate the relationship between educational attainment and AMI survival. Although I applaud the authors' use of population data, I have some additional questions about the methods presented and ask for a bit more clarification.

Main comments

The authors utilize Poisson regression and cubic spline curve to predict survival estimates for 2011-2013 (p. 7). Since the authors have population data from years prior, why was a Bayesian smoothing technique not considered to generate these data?

On a related note, why did the authors decide on using a 6-year follow-up period, especially when data are not available past 2010? Is this related to mean AMI survival periods? Why is 2008-2010 used as AMI baseline?

For each of the results and tables presented, the authors discuss differences across survival parameters. However, it seems that they did not conduct any statistical tests for the parameters to determine whether these differences are statistically significant (e.g., is relative survival for the primary educational group actually different from other groups?). Did the authors test across parameters?

I think it would be helpful to stratify all of the Table 1 information by sex as well (replacing Table 1 or adding an additional table). The protective effect of education can vary by sex, as does AMI risk and survival. Also, depending on the results of stratifying Table 1 for men and women, perhaps the authors should also stratify the remaining tables by gender. This would aid in the paper's overall contribution as well as the discussion during which the authors mention how AMI survival differs for men and women (p. 14).

Given that occupational stress (not education) is highly correlated with cardiovascular disease and AMI, I think this paper would benefit from a stronger argument for the use of education instead of occupation.

Did the authors consider adjusting survival for experiencing multiple AMI, as this might influence survival and preventive subsequent AMI health behaviors may vary by education level.

Minor comment

Last sentence in Results section of Abstract is somewhat confusing.

Disclaimer: I was asked to review the methods of this paper. Therefore, I have not reviewed any conceptual and/or theoretical component of this paper.

VERSION 1 – AUTHOR RESPONSE

Author response - Version 1

We wish to thank the reviewers for the constructive comments and BMJ open for the opportunity to revise and resubmit our manuscript.

Editorial Requirements

1. We have added a sub-title describing the study design.
2. The mentioned sections have been removed.

Response to reviewer #1, Sean Randall's comments:

Thank you for the positive remark.

Response to reviewer #2, Corrado De Vito's comments:

We have revised the methods section to make it clearer. In the section describing statistical analyses, the procedure for obtaining the expected survival has rewritten. Also the related supplementary descriptions (S1 and S2) have been modified.

Response to reviewer #3, Isidora Vujcic's comments:

1. We have elaborated more on this issue in the discussion, in the sixth paragraph ("We used attained..."). The mixed pattern could result from random variation or other factors influencing patient survival. Please note that Table 1 has been modified, in response to reviewer #5's comments.
2. We have offered an interpretation for why expected survival varies by education in the same paragraph as the previous point.
3. We saw this in the context of point 1 above. We believe that education affects AMI survival through several channels, for instance because patients with higher education get better treatment, as suggested. The type of descriptive methods applied in our study are not so well suited for handling many factors simultaneously. It would seem more appropriate use models such as Cox regression to pursue this issue.

Response to reviewer #4, Annette Kjær Ersbøll's comments:

1. In the methods section (Data 2nd paragraph), we have made it more explicit that we model the first AMI in the observation period and that we do not have information of AMI episodes prior to the observation period. We have also added this as a limitation in the discussion section.
2. Only patients with AMI as main diagnosis were included. The first sentence of the Methods section has been revised.
3. Survival was measured from the date of hospital admission for AMI (Patient Registry, 2008-2010) and until time of death (Causes of Death Registry, 2008-2013). We have tried to clarify this in the Data section.
4. We use data from the Norwegian Patient Registry (2008-2010), and in this period the patients (anonymized) IDs can be linked to other registries. It is correct that we do not have information on AMI episodes prior to 2008 (confer point 1 above).
5. It is correct that the patient survival and the expected survival are proportions. When referring to our results, we have removed "rate" throughout and now generally refer only to "survival".

Response to reviewer #5, Patricia Morton's comments:

1. We have revised the text describing the estimation of expected survival, including the use of Poisson regressions. Bayesian smoothing techniques were not considered because the use of Poisson regressions and restricted cubic splines is quite common in this branch of the literature (confer Royston & Lambert (2011) "Flexible parametric survival analysis using Stata: beyond the Cox model"). We do not claim this method is optimal, but the sensitivity analyses already performed suggest that the results are quite robust.
2. In principle, our observation period, both for patient data (2008-2010) and date of death (2008-

2013), could have been expanded because all our data are obtained from registries. However, due to strict Norwegian privacy policies, extracting and linking patient data is both highly costly and time consuming. Thus, we have relied on all data available from the EuroHOPE project (confer the Acknowledgements section). Patients who are alive at the end of 2013, but who entered, say, in 2010 are considered as censored in the estimations. This means that they influence the estimated cumulative patient survival at 6 years, even though they were observed for a shorter period.

3. To the best of our knowledge, no statistical test suitable for this application has been developed. We have included confidence intervals in all tables. In tables 2 and 3, the confidence intervals are non-overlapping in almost all rows (except for Table 2; after 1 month, secondary and tertiary education). Thus, it seems clear that a null hypothesis assuming equal means across the three educational groups would be rejected.

4. We have revised Table 1, as suggested. For Tables 2 and 3, and the graph, we would prefer to keep the original, aggregated results. Some of the confidence intervals in the new version of Table 1 are relatively wide due to few observations in some strata. The practice of pooling the sexes is not uncommon in the related literature, confer for instance Velders et al. (2014, #20 in our reference list) who did this (they stratified by age).

5. Data on occupation, corresponding to our registry based data on education, are not available to us. In the new version we mention occupation in the discussion, in the sixth paragraph (“We used attained...”), we also refer to studies where occupation and other factors have been studied.

6. We do not have access to information regarding AMI episodes prior to our observation period (2008-2010 for patient data). We have modified the Data section and mentioned this as a limitation in the Discussion.

7. The sentence has been revised.

VERSION 2 – REVIEW

REVIEWER	Isidora Vujcic Institute of Epidemiology Faculty of Medicine University of Belgrade
REVIEW RETURNED	27-Jun-2017

GENERAL COMMENTS	The article has improved considerably
---------------------------------------